# Hyperuricemia and Its Association with Osteoporosis in a Large Asian Cohort

**DOI:** 10.3390/nu14112206

**Published:** 2022-05-26

**Authors:** Jhong-You Li, Jia-In Lee, Cheng-Chang Lu, Yu-De Su, Chao-Tse Chiu, Szu-Chia Chen, Jiun-Hung Geng, Chung-Hwan Chen

**Affiliations:** 1Department of Orthopedics, Kaohsiung Municipal Siaogang Hospital, Kaohsiung Medical University, Kaohsiung 81267, Taiwan; mark163333@gmail.com (J.-Y.L.); cclu0880330@gmail.com (C.-C.L.); u9401112@gmail.com (Y.-D.S.); konstantinechiu@gmail.com (C.-T.C.); 2Department of Orthopedics, Kaohsiung Medical University Hospital, Kaohsiung Medical University, Kaohsiung 80701, Taiwan; 3Graduate Institute of Clinical Medicine, College of Medicine, Kaohsiung Medical University, Kaohsiung 80701, Taiwan; 4Department of Psychiatry, Kaohsiung Medical University Hospital, Kaohsiung Medical University, Kaohsiung 80701, Taiwan; u9400039@gmail.com; 5Department of Orthopedics, College of Medicine, Kaohsiung Medical University, Kaohsiung 80701, Taiwan; 6Department of Internal Medicine, Kaohsiung Municipal Siaogang Hospital, Kaohsiung Medical University, Kaohsiung 81267, Taiwan; scarchenone@yahoo.com.tw; 7Division of Nephrology, Department of Internal Medicine, Kaohsiung Medical University Hospital, Kaohsiung Medical University, Kaohsiung 80701, Taiwan; 8Faculty of Medicine, College of Medicine, Kaohsiung Medical University, Kaohsiung 80701, Taiwan; 9Research Center for Environmental Medicine, Kaohsiung Medical University, Kaohsiung 80701, Taiwan; 10Department of Urology, Kaohsiung Municipal Siaogang Hospital, Kaohsiung Medical University, Kaohsiung 81267, Taiwan; 11Department of Urology, Kaohsiung Medical University Hospital, Kaohsiung Medical University, Kaohsiung 80701, Taiwan; 12Orthopaedic Research Center, Kaohsiung Medical University, Kaohsiung 80701, Taiwan; 13Regeneration Medicine and Cell Therapy Research Center, Kaohsiung Medical University, Kaohsiung 80701, Taiwan; 14Department of Orthopedics, Kaohsiung Municipal Ta-Tung Hospital, Kaohsiung City 80145, Taiwan; 15Department of Healthcare Administration and Medical Informatics, Kaohsiung Medical University, Kaohsiung 80701, Taiwan; 16Institute of Medical Science and Technology, National Sun Yat-Sen University, Kaohsiung 80420, Taiwan; 17Graduate Institute of Animal Vaccine Technology, College of Veterinary Medicine, National Pingtung University of Science and Technology, Pingtung 912301, Taiwan; 18Ph.D. Program in Biomedical Engineering, College of Medicine, Kaohsiung Medical University, Kaohsiung 80701, Taiwan

**Keywords:** epidemiologic study, osteoporosis, hyperuricemia, risk factors

## Abstract

In this paper, we aimed to examine the protective role of hyperuricemia in the prevalence of osteoporosis in a large Asian cohort. A total of 119,037 participants from 29 recruitment centers in Taiwan were enrolled onto our study. Participants with serum uric acid greater than 7.0 mg/dL in men and 6.0 mg/dL in women were classified as the hyperuricemia group whereas the others were the control group. The mean age of all participants was 50; there were 23,114 subjects (19%) with hyperuricemia. Osteoporosis was observed in 8243 (9%) and 1871 (8%) participants in the control and hyperuricemia groups, respectively. After adjusting for confounders, a lower risk of osteoporosis was found in the hyperuricemia group compared with the control group (odds ratio, 0.916; 95% confidence interval, 0.864 to 0.970). A subgroup analysis showed that hyperuricemia was associated with a lower risk of osteoporosis in females, but not in males. Women with serum uric acid greater than 8.0 mg/dL were not associated with a greater risk of osteoporosis. Our study suggests that hyperuricemia decreases the risk of osteoporosis in females, but not in males. The protective role was no longer apparent when the serum uric acid level was greater than 8 mg/dL.

## 1. Introduction

Osteoporosis is the most common chronic metabolic disease of the skeletal system and is characterized by a decreased mineral density and the microarchitectural deterioration of the bones [1,2]. The increased fragility of the bones and subsequent osteoporotic fractures such as hip fractures or vertebral compression fractures lead to acute pain, long-term immobilization and even a bedridden status, all of which may be a heavy burden to the public health system. A higher incidence of hip fractures in Taiwan compared with the American white population has been estimated through the nationwide health insurance database [3]. The absolute hip fracture rates in Taiwan are still rising despite a shortened length of hospital stay under the modern healthcare system [4]. Hence, it is crucial to identify factors that minimize the risk of osteoporosis and enhance bone health. Although adequate calcium and vitamin D supplements along with endurance training can help with bone mass [5,6], the osteoprotective properties involved in bone anabolism and catabolism remain largely unknown. In recent studies, oxidative stress reduced bone density through the modulation of osteoclast and osteoblast activities [7,8,9]. Antioxidants may, therefore, act as an osteoprotective agent to slow down bone absorption and osteoporosis.

Uric acid is considered to be a metabolic end product of purine. Elevated serum uric acid (SUA), also known as hyperuricemia, is caused by urate overproduction (a purine-rich diet, errors in purine metabolism and cell breakdown) or decreased uric acid excretion (acute or chronic kidney disease, acidosis, alcohol consumption, diuretics, lead intoxication, hyperparathyroidism and hypothyroidism) [10,11,12]. Hyperuricemia is associated with several cardiovascular and metabolic diseases such as coronary artery disease, heart failure, gouty arthritis and metabolic syndrome [13,14,15]. However, there is growing evidence that hyperuricemia is associated with a decrease in the prevalence of osteoporosis [16,17,18]. Potential mechanisms may relate to SUA as a powerful antioxidant both in vitro and in vivo [19,20]. Local tissue damage and a reduced bone density may partly mediate the presence of free radicals such as singlet oxygen, peroxyl radicals and hydroxyl radicals. Elevated uric acid with reduced oxidative stress in vivo may contribute to its protective role in osteoporosis.

In this study, we aimed to evaluate the association between hyperuricemia and osteoporosis after correcting for confounding factors in a large Asian cohort. We hypothesized that hyperuricemia is negatively associated with the prevalence of osteoporosis and that there is a cut-off value for SUA with regard to osteoporosis.

## 2. Materials and Methods

### 2.1. Subjects

We collected subjects from the Taiwan Biobank (TWB) to evaluate the association between hyperuricemia and the risk of osteoporosis. Details of the TWB have been previously described [21,22]. Briefly, the TWB enrolled more than 100,000 volunteers from 29 recruitment centers across the country. This included information on clinical characteristics, blood tests and bone mineral density (BMD). We excluded participants with inadequate data regarding gender, age, body mass index (BMI) and SUA. In total, there were 119,037 subjects enrolled in the final analysis (Figure 1).

### 2.2. Ethics Statement

Our study was approved by the Institutional Review Board of Kaohsiung Medical University Hospital (KMUHIRB-E(I)-20210058). All participants signed written informed consent and the investigators followed the Declaration of Helsinki.

### 2.3. Definition of Hyperuricemia

All participants had blood tests, including for SUA. Participants with SUA greater than 7.0 mg/dL in men and 6.0 mg/dL in women were classified as the hyperuricemia group whereas the others were the control group.

### 2.4. Definition of Osteoporosis

The estimated BMD (g/cm^2^) was evaluated by using ultrasound (Achilles InSight, GE, Madison Heights, USA) at the heel calcaneus [23]. The formula used was T score = (individual’s BMD—mean BMD in young adults)/standard deviation (SD) of a normal young adult population. The presence of osteoporosis was defined as a T score ≤ −2.5 SD below the young adult level.

### 2.5. Covariates

A directed acyclic graph was used to identify the possible confounders of the relationship between hyperuricemia and osteoporosis (Appendix A), including age [24,25], gender [24,25], the presence of menopause [24], smoking [24], drinking [1,26], obesity [1,25,26], diet [24,25], physical activity [1], hypertension [25,27], diabetes mellitus [1], dyslipidemia [28] and chronic kidney disease [10]. The directed acyclic graph was drawn using the program daggity [29,30]. These covariates were collected from self-reported questionnaires supplemented by physical examinations or blood tests.

### 2.6. Statistical Analyses

We divided the participants into two groups: the control group and the hyperuricemia group. The mean ± SD and percentages were used to describe the continuous variables and the categorical variables in the general clinical profiles of the participants, respectively. Chi-squared tests and independent *t*-tests were used to compare the categorical variables and the continuous variables within the groups, respectively. A logistic regression was conducted to examine the association between the presence of hyperuricemia and osteoporosis. We also conducted a multivariable linear regression to analyze the association between SUA and the estimated BMD. R (version 3.6.2, R Foundation for Statistical Computing, Wien, Austria and SPSS (version 20.0, IBM Corp, Armonk, NY, USA) were the tools used for performing all analyses. A *p*-value < 0.05 represented statistically significant results.

## 3. Results

### 3.1. Comparison of the Baseline Clinical Profiles among the Participants According to the Presence of Hyperuricemia

A total of 119,037 subjects were enrolled into our study with a mean age of 50 ± 11 years old. There were 23,114 subjects (19%) with hyperuricemia (Table 1). The participants in the hyperuricemia group tended to be older and of the male gender as well as with the presence of menopause, a higher BMI, higher smoking and drinking rates and higher blood pressure, with a higher prevalence of hypertension, diabetes mellitus (DM), dyslipidemia, gout, total cholesterol, triglycerides, creatinine and higher serum fasting glucose, but lower T scores than those in the control group (Table 1).

### 3.2. Association between the Presence of Hyperuricemia and Osteoporosis

Out of all the participants, 10,114 subjects (9%) had osteoporosis; 8243 participants (9%) in the control group and 1871 (8%) in the hyperuricemia group (Table 1). The subjects with a higher BMI and higher serum albumin levels had lower odds of osteoporosis (Table 2). The subjects in the hyperuricemia group were associated with a lower prevalence of osteoporosis compared with those in the control group (odds ratio (OR), 0.937; 95% confidence interval (95% CI), 0.889 to 0.987) (Table 2). In the multivariable logistic regression analysis, with adjustments for the variables (Table 2), the participants in the hyperuricemia group were significantly associated with a lower risk of osteoporosis than those in the control group (OR, 0.916; 95% CI, 0.864 to 0.970) (Table 3). A multivariable linear regression analysis also showed a positive association between SUA and the estimated BMD both in all subjects and in those participants in the control group (Appendix A). The results were similar in the analyses after excluding those subjects with DM or a BMI ≥ 30 kg/m^2^ (Appendix A).

### 3.3. Association between the Presence of Hyperuricemia and Osteoporosis in participants Stratified by Age and Sex

A subgroup analysis was conducted to examine the association between hyperuricemia and osteoporosis. Interestingly, after adjusting for confounders, we found that hyperuricemia was associated with a lower risk of osteoporosis in females > 65 years old (OR, 0.806; 95% CI, 0.662 to 0.981) and females ≤ 65 years old (OR, 0.863; 95% CI, 0.785 to 0.950), but not in males (Table 4).

### 3.4. Dose–Response Effect between Serum Uric Acid and the Risk of Osteoporosis

To further examine the dose–response effect between serum uric acid and the risk of osteoporosis, we divided the females by their serum uric acid levels. As shown in Table 5, the adjusted odds of osteoporosis were 0.859 (95% CI, 0.797 to 0.926), 0.824 (95% CI, 0.757 to 0.896), 0.794 (95% CI, 0.714 to 0.883), 0.641 (95% CI, 0.532 to 0.773) and 0.777 (95% CI, 0.585 to 1.031) compared with the reference for females with serum uric acid levels of 4 to 5 mg/dL, 5 to 6 mg/dL, 6 to 7 mg/dL, 7 to 8 mg/dL and > 8 mg/dL, respectively. There was a decreasing trend in the risk of osteoporosis for serum uric acid levels up to 8 mg/dL, but not for those with > 8 mg/dL in females.

## 4. Discussion

In this cross-sectional study of a representative population in Taiwan, hyperuricemia decreased the risk of osteoporosis in females, but not in males after an adjustment for confounders. Furthermore, there was a dose–response association between the levels of serum uric acid and the risk of osteoporosis up to 8 mg/dL, but not for those >8 mg/dL.

Evidence from studies has linked a reduced BMD to oxidative stress. The antioxidative effect of SUA could also contribute to its protective role in osteoporosis. Despite the well-documented association between hyperuricemia and BMD, there are inconsistent study designs and statistical findings. Yao et al. found a positive correlation between SUA levels and lumbar BMD in both women and men [16]. However, when further stratified by ethnicity, a positive correlation was not found in black people; the association between the SUA level and the BMD was an inverted U shape with a paradoxical lower BMD after SUA was >7.5 mg/dL. Yan et al. used BMD and bone turnover markers such as type 1 N-terminal propeptide (P1NP) to evaluate bone density and metabolism in a Chinese population [17]. Significant correlations were found between the SUA levels and the BMD or SUA levels and P1NP in postmenopausal women only. Few studies have analyzed bone health in older men in association with SUA levels. Although an article published in the *Journal of Bone and Mineral Research* (*JBMR*) in 2011 by Nabipour et al. [31] revealed higher SUA levels associated with a higher BMD in older men at all skeletal sites, another study conducted by Lane et al. in 2014 revealed no relationship between the SUA levels and a higher hip BMD or hip fractures; there were, however, associations with a reduction in the risk of non-spine fractures [32].

Despite diversity in the patient selection and confounding factors compared with previous studies, most of the studies disclosed significant associations between higher SUA levels and a higher BMD. Whether sex or ethnicity played an important role in this association remains unclear. The antioxidative effect of uric acid may be different between postmenopausal women and older men based on their different pathophysiologies. We believe that unlimited higher SUA levels do not equal an unlimited higher BMD. In our study, there was a decreasing trend of incidences of osteoporosis for SUA levels up to 8 mg/dL with a dose effect, but not for those above 8 mg/dL in females. Our results suggested that there was a cut-off value for the SUA levels (8 mg/dL), with no other osteoprotective effect noted thereafter.

There were several limitations to this study. First, quantitative ultrasonography (QUS) was used to calculate the heel calcaneus BMD measurements in our study. The majority of studies nowadays use dual energy X-ray absorptiometry (DXA) as a standard measurement. However, ultrasound is an alternative measurement tool that is safe, quick and relatively inexpensive, making it a perfect tool for a national databank analysis. Studies have proven the correlation between ultrasound BMD and DXA BMD [23]; BMD derived from ultrasound independently predicted the risk of a hip fracture and non-spine fractures [33,34]. Second, due to the design of the cross-sectional study, it was difficult to establish causality between BMD and SUA. Basic in vitro or in vivo studies are needed to elucidate the mechanism behind our results. Third, there were no data on the medical record of fragile fractures in this study, which limited our understanding of the effects of hyperuricemia on the decrease in risk of fragile fractures. Fourth, all the subjects were Asian, so it was not possible to verify our findings across ethnicities. Fifth, despite several confounding factors such as the BMI, common comorbidities and laboratory data being adjusted, we did not evaluate lifestyle factors such as dietary calcium and vitamin D supplementation, which can affect the BMD. The treatment of patients with chronic diseases (e.g., patients on chemotherapy for cancer or long-term steroid use) could interfere with the relationship between the BMD and SUA. Finally, the non-significant *p*-value for high SUA levels (Table 5) could be due to a lower power in this group (of only 63 cases). The possible upper limit of the protective effect of SUA on osteoporosis needs further exploration. All of these limitations certainly make our findings inconclusive and warrant further research to better understand the causal effects between these two factors.

Our study has a number of strengths. This is the largest study evaluating the association between BMD and SUA in the Asian population. Relatively large numbers of subjects were recruited with a broad spectrum of variables. Differences in gender and age were identified through a subgroup analysis. Osteoporosis and its related osteoporotic fractures are a huge burden on the public health system. Hence, it is necessary to identify the factors associated with bone health and further evaluate these relationships. In our study, we found a possible protective effect of hyperuricemia on osteoporosis, particularly in women, when the SUA levels were below 8 mg/dL.

## 5. Conclusions

In conclusion, our large population-based study demonstrated that hyperuricemia was associated with a decreased risk of osteoporosis in females, but not in males. The protective role was no longer apparent when the SUA levels were greater than 8 mg/dL. Further studies should be conducted to validate our results. We believe that our study offers alternative ideas on osteoporosis prevention and treatment in the long run.

## Figures and Tables

**Figure 1 nutrients-14-02206-f001:**
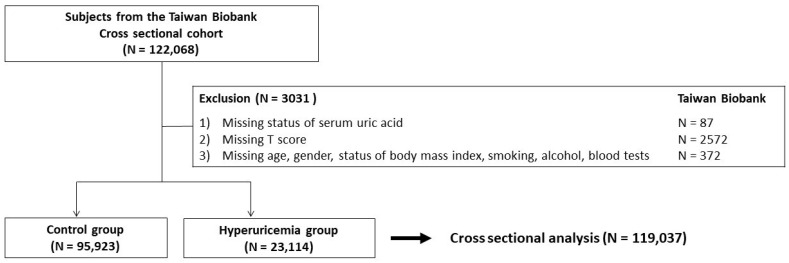
Study participants classified by the presence of hyperuricemia.

**Table 1 nutrients-14-02206-t001:** General clinical profiles of participants (N = 119,037).

Characteristics	Total (N = 119,037)	Control Group (N = 95,923)	Hyperuricemia Group (N = 23,114)	*p*-Value
Demographic data				
Age, years old	50 ± 11	50 ± 11	51 ± 11	<0.001
Female, N (%)	76,313 (64)	65,939 (69)	10,374 (45)	<0.001
Body mass index, kg/m^2^	24.2 ± 3.8	23.7 ± 3.5	26.5 ± 3.9	<0.001
Smoked, ever, N (%)	10,139 (9)	23,738 (25)	8700 (38)	<0.001
Alcohol status, ever, N (%)	715 (2)	6922 (7)	3217 (14)	<0.001
Regular physical activity, yes, N (%)	48,351 (41)	38,680 (40)	9671 (42)	<0.001
Presence of menopause, yes, N (%) *	34,985 (46) *	28,626 (43) *	6359 (61) *	<0.001
Systolic blood pressure, mmHg	121 ± 19	119 ± 18	127 ± 18	<0.001
Diastolic blood pressure, mmHg	74 ± 11	73 ± 11	78 ± 11	<0.001
Comorbidities				
Hypertension, N (%)	14,596 (12)	9759 (10)	4837 (21)	<0.001
Diabetes mellitus, N (%)	6128 (5)	4664 (5)	1464 (6)	<0.001
Dyslipidemia, N (%)	8880 (8)	6376 (7)	2504 (11)	<0.001
Gout, N (%)	4558 (4)	1663 (2)	2895 (13)	<0.001
Laboratory data				
Albumin, g/dL	4.5 ± 0.2	4.5 ± 0.2	4.6 ± 0.2	<0.001
Fasting glucose, mg/dL	96 ± 21	95 ± 21	98 ± 19	<0.001
Total cholesterol, mg/dL	196 ± 36	194 ± 35	202 ± 37	<0.001
Triglyceride, mg/dL	116 ± 94	105 ± 81	158 ± 127	<0.001
High-density lipoprotein cholesterol, mg/dL	55 ± 13	56 ± 13	48 ± 12	<0.001
Low-density lipoprotein cholesterol, mg/dL	121 ± 32	119 ± 31	128 ± 33	<0.001
Creatinine, mg/dL	0.7 ± 0.3	0.7 ± 0.3	0.9 ± 0.4	<0.001
T score	−0.387 ± 1.627	−0.369 ± 1.640	−0.460 ± 1.570	<0.001
Osteoporosis, N (%)	10,114 (9)	8243 (9)	1871 (8)	0.015

* Presence of menopause was only counted in women.

**Table 2 nutrients-14-02206-t002:** Parameters associated with osteoporosis in univariate binary logistic analysis in all study participants (N = 119,037).

Parameters	Odds Ratio (95% CI)	*p*-Value
Age (per 1 year)	1.078 (1.076 to 1.081)	<0.001
Gender, female (vs. male)	0.783 (0.751 to 0.816)	<0.001
Body mass index (per 1 kg/m^2^)	0.952 (0.946 to 0.957)	<0.001
Smoking status, ever (vs. never)	1.161 (1.110 to 1.214)	<0.001
Alcohol status, ever (vs. never)	1.195 (1.116 to 1.280)	<0.001
Regular physical activity, yes (vs. no)	1.298 (1.246 to 1.352)	<0.001
Presence of menopause, yes (vs. no)	0.937 (0.889 to 0.987)	0.015
Systolic blood pressure (per 1 mmHg)	1.013 (1.012 to 1.014)	<0.001
Diastolic blood pressure (per 1 mmHg)	1.007 (1.006 to 1.009)	<0.001
Hypertension, yes (vs. no)	1.538 (1.456 to 1.625)	<0.001
Diabetes mellitus, yes (vs. no)	1.435 (1.323 to 1.556)	<0.001
Dyslipidemia, yes (vs. no)	1.439 (1.343 to 1.541)	<0.001
Gout, yes (vs. no)	1.145 (1.035 to 1.267)	0.008
Albumin (per 1 g/dL)	0.649 (0.595 to 0.708)	<0.001
Fasting glucose (per 1 g/dL)	1.005 (1.004 to 1.006)	<0.001
Total cholesterol (per 1 mg/dL)	1.003 (1.002 to 1.003)	<0.001
Triglyceride (per 1 mg/dL)	1.000 (1.000 to 1.001)	<0.001
High-density lipoprotein cholesterol (per 1 mg/dL)	1.002 (1.001 to 1.004)	0.006
Low-density lipoprotein cholesterol (per 1 mg/dL)	1.002 (1.001 to 1.002)	<0.001
Creatinine, mg/dL	1.236 (1.181 to 1.293)	<0.001
Hyperuricemia, yes (vs. control)	0.937 (0.889 to 0.987)	0.015

CI: confidence interval.

**Table 3 nutrients-14-02206-t003:** Parameters associated with osteoporosis in multivariable binary logistic analysis in all study participants (N = 119,037).

Parameters	Odds Ratio (95% CI)	*p*-Value
Age (per 1 year)	1.067 (1.064 to 1.070)	<0.001
Gender, female (vs. male)	0.409 (0.376 to 0.444)	<0.001
Body mass index (per 1 kg/m^2^)	0.921 (0.915 to 0.928)	<0.001
Smoking status, ever (vs. never)	1.150 (1.085 to 1.218)	<0.001
Alcohol status, ever (vs. never)	1.029 (0.953 to 1.111)	0.465
Regular physical activity, yes (vs. no)	0.726 (0.694 to 0.759)	<0.001
Presence of menopause, yes (vs. no)	2.375 (2.202 to 2.561)	<0.001
Systolic blood pressure (per 1 mmHg)	1.002 (1.000 to 1.004)	0.027
Diastolic blood pressure (per 1 mmHg)	0.999 (0.996 to 1.001)	0.330
Hypertension, yes (vs. no)	1.024 (0.962 to 1.091)	0.453
Diabetes mellitus, yes (vs. no)	0.912 (0.828 to 1.006)	0.065
Dyslipidemia, yes (vs. no)	0.957 (0.889 to 1.031)	0.247
Gout, yes (vs. no)	0.950 (0.852 to 1.060)	0.359
Albumin (per 1 g/dL)	0.850 (0.771 to 0.937)	0.001
Fasting glucose (per 1 g/dL)	1.000 (0.999 to 1.001)	0.481
Total cholesterol (per 1 mg/dL)	0.998 (0.996 to 1.000)	0.091
Triglyceride (per 1 mg/dL)	1.001 (1.000 to 1.001)	<0.001
High-density lipoprotein cholesterol (per 1 mg/dL)	1.003 (1.000 to 1.006)	0.072
Low-density lipoprotein cholesterol (per 1 mg/dL)	1.002 (0.999 to 1.004)	0.194
Creatinine, mg/dL	1.061 (0.999 to 1.127)	0.054
Hyperuricemia, yes (vs. control)	0.916 (0.864 to 0.970)	0.003

CI: confidence interval. Multivariable model: adjustment for age, gender, body mass index, smoking status, alcohol status, regular physical activity, presence of menopause, systolic blood pressure, diastolic blood pressure, history of hypertension, history of diabetes mellitus, history of dyslipidemia, history of gout, fasting glucose, cholesterol, triglyceride, low-density lipoprotein cholesterol, high-density lipoprotein cholesterol, serum albumin and creatinine.

**Table 4 nutrients-14-02206-t004:** Univariate and multivariable binary logistic analysis for the prevalence of osteoporosis in subgroup analyses.

Characteristics	Crude Odds Ratio (95% CI)	*p*-Value	Adjusted Odds Ratio (95% CI)	*p*-Value
Gender, female, >65 years old (N = 4757)				
Hyperuricemia (+)	0.641 (0.536 to 0.767)	<0.001	0.806 (0.662 to 0.981)	0.032
Hyperuricemia (−)	1.00		1.00	
Gender, female, ≤65 years old (N = 71,556)				
Hyperuricemia (+)	0.988 (0.906 to 1.077)	0.779	0.863 (0.785 to 0.950)	0.003
Hyperuricemia (−)	1.00		1.00	
Gender, male, >65 years old (N = 3763)				
Hyperuricemia (+)	0.863 (0.701 to 1.062)	0.164	0.966 (0.772 to 1.208)	0.762
Hyperuricemia (−)	1.00		1.00	
Gender, male, ≤65 years old (N = 38,961)				
Hyperuricemia (+)	0.797 (0.737 to 0.862)	<0.001	0.932 (0.856 to 1.015)	0.106
Hyperuricemia (−)	1.00		1.00	

CI: confidence interval. Multivariable model: adjustment for age, gender, body mass index, smoking status, alcohol status, regular physical activity, presence of menopause, systolic blood pressure, diastolic blood pressure, history of hypertension, history of diabetes mellitus, history of dyslipidemia, history of gout, fasting glucose, cholesterol, triglyceride, low-density lipoprotein cholesterol, high-density lipoprotein cholesterol, serum albumin and creatinine.

**Table 5 nutrients-14-02206-t005:** Odds ratio for the prevalence of osteoporosis according to the levels of serum uric acid in females (N = 76,313).

Characteristics	Number of Cases (%)	Number at Risk	Adjusted Odds Ratio (95% CI)	*p*-Value
Serum uric acid ≤ 4.0 mg/dL	1387 (7.8)	17,819	1.00 (reference)	
4.0 mg/dL < serum uric acid ≤ 5.0 mg/dL	2168 (7.6)	28,516	0.859 (0.797 to 0.926)	<0.001
5.0 mg/dL < serum uric acid ≤ 6.0 mg/dL	1467 (8.0)	18,263	0.824 (0.757 to 0.896)	<0.001
6.0 mg/dL < serum uric acid ≤ 7.0 mg/dL	713 (8.1)	8789	0.794 (0.714 to 0.883)	<0.001
7.0 mg/dL < serum uric acid ≤ 8.0 mg/dL	152 (7.0)	2166	0.641 (0.532 to 0.773)	<0.001
Serum uric acid > 8.0 mg/dL	63 (8.3)	760	0.777 (0.585 to 1.031)	0.081

CI: confidence interval. Multivariable model: adjustment for age, body mass index, smoking status, alcohol status, regular physical activity, presence of menopause, systolic blood pressure, diastolic blood pressure, history of hypertension, history of diabetes mellitus, history of dyslipidemia, history of gout, fasting glucose, cholesterol, triglyceride, low-density lipoprotein cholesterol, high-density lipoprotein cholesterol, serum albumin and creatinine.

## Data Availability

The data underlying this study are from the Taiwan Biobank. Due to restrictions placed on the data by the Personal Information Protection Act of Taiwan, the minimal data set cannot be made publicly available. Data may be available upon request to interested researchers. Please send data requests to Szu-Chia Chen, Division of Nephrology, Department of Internal Medicine, Kaohsiung Medical University Hospital, Kaohsiung Medical University.

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
