# Peer review of "Hyperuricemia and Its Association with Osteoporosis in a Large Asian Cohort"

_nutrients, 2022, doi:10.3390/nu14112206_

Round 1

Reviewer 1 Report

In this cross-sectional study of a representative population in Taiwan, the authors aimed to explore the protective role of hyperuricemia on the prevalence of osteoporosis. They recruited a total of 119,037 participants and considered serum uric acid greater than 7.0 mg/dL in men and 6.0 mg/dL in women as indicative of the hyperuricemia. Bone mineral density was assessed by quantitative calcaneal ultrasonometry. Their results were consistent with a decreased risk of osteoporosis in females, but not in males, with a dose-response association between serum uric acid levels and risk of osteoporosis.

The main finding of the study is not really new, as serum uric acid has been previously correlated with BMD.

The authors may also consider analyzing continuous data from quantitative ultrasound measurements and then correlating them with uric acid concentrations.

Is there a correlation between BMD and uric acid even in the control group with normal uric acid concentrations?

What about these correlations after exclusion of obese and/or diabetic participants? It is known that these diseases may be associated with higher BMD.

What about fragility fractures?

While no correction for known risk factors for osteoporosis (and fractures) does not reduce the relevance of this study, it certainly makes the result on serum uric acid inconclusive.

Author Response

Please find the attached about our responses, thanks. 

Reviewer 2 Report

Review: Hyperuricemia and Its Association With Osteoporosis in A 2 Large Asian Cohort

This is a well-written and easy to understand study of a topic that has not been much investigated. The results are nicely presented in tables. However, I have concerns about the methods used, which are a somewhat outdated and not well described. I summarize my comments below.

Introduction/background:

The introduction/background is to the point and well written. Minor edits, such as “A higher incidence” instead of “the higher incidence” on line 48, and “absolute hip fracture rates” on line 50 should be made. Please also use the words “relation” or “association” instead of “relationship”. I would also like more information about hyperuricemia (why does it occur in some people?).

Methods:

The selection of participants is nicely illustrated in a flow-chart.

More description of possible confounding variables and the selection of confounding variables is needed. Do not put too much emphasis on p-values, with such a large sample almost everything will be significant. Rather than using a data-driven method for the selection of possible confounders, draw a Directed Acyclic Graph (DAG) of the proposed causal relation between hyperuricemia and osteoporosis (for example by using the program daggity: http://www.dagitty.net/). A DAG will give you a minimal sufficient adjustment set. You can also include the unknown variables, such as lifestyle factors in the DAG. It is important to avoid overadjustment (rather use the minimal sufficient adjustment set), as overadjustment can cause spurious significant associations.

Results:

The results are well described; however, they must be changed according to the necessary changes in the Methods (above).

Some minor errors in table 1, where the thousand separator is not included: 76313 and 10374.

What is the p-value for total cholesterol in table 2?

Do not use too many decimal points in tables and text. One or two is sufficient.

Again, do not put too much emphasis on p-values, p-values are not necessary to include in descriptive tables. The effect estimate (odds ratio) is what is important and should be interpreted. The non-significant p-value for high serum uric acid level (table 5) could simply be due to lower power in this group (only 63 cases).

Discussion:

Again, do not put too much emphasis on the p-value of the upper limit. It may be due to low power. Look at the estimate!

What is a cross-sectional cohort study? Line 195. If you have no follow-up it is a cross-sectional study, if follow-up it is a cohort study.

Do you have a reference to the validity of QUS?

Author Response

Please find attached about our responses, thanks. 

Round 2

Reviewer 1 Report

The manuscript has been improved after revision

Author Response

Thank you for giving us the opportunity to strengthen our manuscript with your valuable comments and queries.

Sincerely yours,

JIUN-HUNG GENG; Chung-Hwan Chen

Reviewer 2 Report

Dear Authors. Thank you for your satisfactory response to my queries. One minor comment: remember the difference between multivariable regression models (several independent variables) and multivariate regression models (several dependent variables), please use the appropriate term. Other than that I have no more comments. 

Author Response

Dear reviewers:

Thank you for your precious comments. We revised the statistical method to be "multivariable" regression models. The corresponding descriptions are as follows: “ We also conducted a multivariable linear regression to analyze the association between SUA and estimated BMD." (Line 123); also line 146; line 150; table 3; table 4; Supplementary Table 1.

Sincerely yours,

JIUN-HUNG GENG; Chung-Hwan Chen